# Distinguishing faceted oxide nanocrystals with $^{17}$O solid-state NMR spectroscopy

Yuhong Li[1,2], Xin-Ping Wu [3], Ningxin Jiang[1], Ming Lin[4], Li Shen[1], Haicheng Sun[1], Yongzheng Wang[1], Meng Wang[1], Xiaokang Ke[1], Zhiwu Yu[5], Fei Gao[1,6], Lin Dong[1,6], Xuefeng Guo[1], Wenhua Hou[1], Weiping Ding[1], Xue-Qing Gong[3], Clare P. Grey[7,8] & Luming Peng[1]

Facet engineering of oxide nanocrystals represents a powerful method for generating diverse properties for practical and innovative applications. Therefore, it is crucial to determine the nature of the exposed facets of oxides in order to develop the facet/morphology–property relationships and rationally design nanostructures with desired properties. Despite the extensive applications of electron microscopy for visualizing the facet structure of nanocrystals, the volumes sampled by such techniques are very small and may not be representative of the whole sample. Here, we develop a convenient $^{17}$O nuclear magnetic resonance (NMR) strategy to distinguish oxide nanocrystals exposing different facets. In combination with density functional theory calculations, we show that the oxygen ions on the exposed (001) and (101) facets of anatase titania nanocrystals have distinct $^{17}$O NMR shifts, which are sensitive to surface reconstruction and the nature of the steps on the surface. The results presented here open up methods for characterizing faceted nanocrystalline oxides and related materials.

---

[1] Key Laboratory of Mesoscopic Chemistry of Ministry of Education and Collaborative Innovation Center of Chemistry for Life Sciences, School of Chemistry and Chemical Engineering, Nanjing University, Nanjing 210023, China. [2] Jiangsu Laboratory of Advanced Functional Materials, School of Chemistry and Material Engineering, Changshu Institute of Technology, Changshu 215500, China. [3] Key Laboratory for Advanced Materials, Centre for Computational Chemistry and Research Institute of Industrial Catalysis, School of Chemistry & Molecular Engineering, East China University of Science and Technology, Shanghai 200237, China. [4] Institute of Materials Research and Engineering, A*STAR (Agency for Science, Technology and Research), 2 Fusionopolis Way, #08-03, Innovis, Singapore 138634, Republic of Singapore. [5] High Magnetic Field Laboratory of the Chinese Academy of Sciences, Hefei 230031, China. [6] Jiangsu Key Laboratory of Vehicle Emissions Control, Center of Modern Analysis, Nanjing University, Nanjing 210093, China. [7] Department of Chemistry, University of Cambridge, Lensfield Road, Cambridge CB2 1EW, UK. [8] Department of Chemistry, Stony Brook University, Stony Brook, NY 11974-3400, USA. Yuhong Li and Xin-Ping Wu contributed equally to this work. Correspondence and requests for materials should be addressed to X.-Q.G. (email: xgong@ecust.edu.cn) or to L.P. (email: luming@nju.edu.cn)

Faceted oxide nanocrystals have attracted much research attention in a variety of fields, including catalysis[1–4], photocatalysis[5–8], solar hydrogen generation[9], photoelectrochemical application[10], gas sensing[11], and energy storage[12], owing to their specific surface structures. Identification of the exposed facets is thus fundamental to the preparation and applications of oxide nanomaterials. Current characterization tools for studying the surface structure of nanocrystals are mostly based on electron microscopy[13–18]. At a resolution that the exposed facet can be determined, however, the field of view of microscopy techniques is often so small, or the particles may show considerable aggregation that it is possible that the region investigated is not representative of the whole sample[19]. Therefore, the development of complementary characterization methods that can give detailed structural information concerning the nature of the exposed facets of nanocrystals is urgently required.

Solid-state NMR spectroscopy is a powerful technique that has been widely used in studying the local environments of solids[20]. $^{17}O$ NMR spectra, e.g., can give detailed structural and dynamic information of important functional oxygen-containing materials[21–26], benefiting from the large $^{17}O$ chemical shift range (>1000 ppm). However, few publications are available on the $^{17}O$ NMR studies of nanosized oxides, in spite of their widespread applications, largely owing to the high cost of $^{17}O$ and structure change during isotopic labeling. Recently, Wang et al.[27] developed a surface-selective labeling method for oxide nanomaterials at low temperatures and revealed that the $^{17}O$ species on the first few layers of ceria nanomaterials are associated with different $^{17}O$ chemical shifts. However, direct experimental evidence is still missing concerning the relationship between the $^{17}O$ chemical shifts and the nature of the exposed facets.

Here, we demonstrate a new approach based on NMR and surface-selective $^{17}O$ labeling to determine the structures of the exposed facets on the technologically important anatase titania nanocrystals[6, 9, 10, 28–30]. With the help of density functional theory (DFT) calculations, oxygen species on different facets can be distinguished by their NMR shifts. The nature of surface steps and reconstructions of these surfaces, particularly on reaction with water, are also revealed.

## Results

**Morphology of anatase TiO$_2$ nanosheets and nano-octahedra.** Two types of anatase TiO$_2$ nanocrystals with different tailored facets were examined, i.e., anatase TiO$_2$ nanosheets with dominant exposed (001) facets (NS001-TiO$_2$), and nano-octahedra preferentially exposing (101) facets (NO101-TiO$_2$)[31]. Their crystal forms were confirmed with X-ray diffraction (XRD) (Supplementary Fig. 1). High-resolution transmission electron microscopy (HRTEM) results show that NS001-TiO$_2$ (Supplementary Fig. 2) are nanosheets with a thickness of 6–7 nm, while NS101-TiO$_2$ (Supplementary Fig. 3) are nano-octahedra with an average size of 14 nm. According to the statistical analysis of the data (Supplementary Figs. 2, 3), an average of 77% of the exposed surfaces of NS001-TiO$_2$ are (001) facets, while 96% of the exposed surfaces of NO101-TiO$_2$ are (101) (see Supplementary Table 1, Supplementary Fig. 4 and additional discussion in Supplementary Note 1). X-ray photoelectron spectroscopy (XPS) spectra (Supplementary Fig. 5) suggest that there is no evidence for the existence of F$^-$ or Cl$^-$ on the surface of either sample, while the concentrations of carbon (C) or nitrogen (N) impurities in both samples are also very small according to the elemental analysis (Supplementary Table 2).

**$^{17}O$ NMR spectra of the surface-selectively labeled samples.** After exposing to $^{17}O$-water for surface-selective labeling[27], the

anatase nanocrystals were characterized with $^{17}O$ magic angle spinning (MAS) NMR spectroscopy and were compared to a non-faceted anatase TiO$_2$ sample with a smaller surface area (denoted as NF1-TiO$_2$, see Supplementary Fig. 6) labeled nonselectively with $^{17}O_2$ at 500 °C, as shown in Fig. 1. Bulk anatase TiO$_2$ consists of TiO$_6$ octahedra that share 4 O–O edges (Supplementary Fig. 7a and Supplementary Table 3) and all of the O ions are 3-coordinated (OTi$_3$, denoted as O$_{3c}$) with an average Ti–O bond length of 0.195 nm[32]. Therefore, $^{17}O$ NMR spectrum of the anatase TiO$_2$ sample enriched with $^{17}O_2$ at high temperature show a single sharp peak at 558 ppm (Fig. 1), corresponding to O$_{3c}$ species in the "bulk" part, consistent with previous reports[33]. It is clear that the $^{17}O$ NMR spectra of surface-labeled NS001-TiO$_2$ and NO101-TiO$_2$ differ significantly and are also distinct from the spectrum of the nonselectively labeled anatase TiO$_2$ (Fig. 1), reflecting their different local structures (see Supplementary Fig. 7 and Supplementary Table 3, and further discussions below), suggesting that $^{17}O$ NMR spectroscopy can be a new method to distinguish faceted oxide nanocrystals.

The signals observed in the $^{17}O$ NMR spectra of NS001-TiO$_2$ and NO101-TiO$_2$ can be categorized into three types. The resonances at 480–570 ppm should arise from O$_{3c}$ species on the surface of titania, since their chemical shifts are close to that of bulk O$_{3c}$. The peaks at higher frequencies (600–750 ppm) can be assigned to O species with lower coordination numbers (e.g., O$_{2c}$)

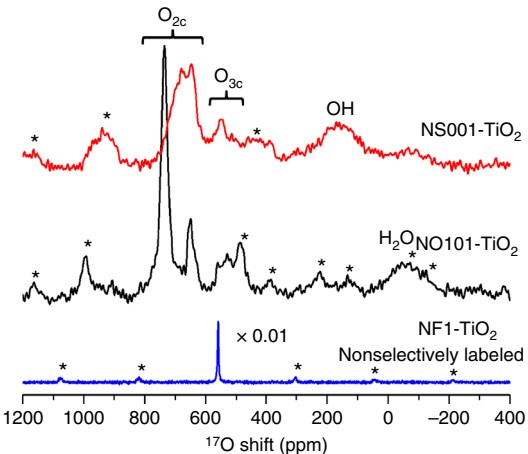

**Fig. 1** $^{17}O$ NMR spectra of faceted anatase titania nanocrystals compared to the non-faceted sample. Anatase TiO$_2$ nanosheets with dominant exposed (001) facets (NS001-TiO$_2$), and nano-octahedra preferentially exposing (101) facets (NO101-TiO$_2$) were surface-selectively $^{17}O$-labeled and vacuum dried for 2 and 12 h, respectively. The other sample, NF1-TiO$_2$, was nonselectively $^{17}O$-labeled. All data were obtained at 9.4 T under a MAS frequency of 14 kHz. A rotor synchronized Hahn-echo sequence ($\pi/6$–$\tau$–$\pi/3$–$\tau$—acquisition) and optimized recycle delays (0.5 s for NS001-TiO$_2$ and NO101-TiO$_2$, and 50 s for NF1-TiO$_2$, see Supplementary Fig. 8), with $^1H$ decoupling, were used to obtain the NMR data. 120,000, 110,000, and 1200 scans were collected for NS001-TiO$_2$, NO101-TiO$_2$, and NF1-TiO$_2$, respectively. The spectra are normalized according to the sample mass and the number of scans (Supplementary Table 4). Asterisks denote sidebands. The dependence of the $^{17}O$ MAS NMR spectra of the two faceted samples on the vacuum-drying time is shown in Supplementary Figs. 9 and 10, and discussed in the Supplementary Notes 2 and 3. Comparison of the $^{17}O$ NMR spectra of the two faceted nanocrystalline samples to a surface-selectively labeled, non-faceted anatase TiO$_2$ nanoparticle sample with comparable surface area (denoted as NF2-TiO$_2$ and the TEM image of the sample is shown in Supplementary Fig. 11) can be found in Supplementary Fig. 12 and Supplementary Note 4

on the surface of titania nanostructure[27]. The broad signals at much lower frequencies (−150 to 300 ppm) can be attributed to hydroxyl groups in surface hydroxyls and/or water environments[27, 34, 35]. The peak centered at 150 ppm in the spectrum of NS001-TiO₂ can also be observed in $^1H \rightarrow ^{17}O$ cross-polarization (CP) MAS NMR spectra (Supplementary Fig. 13 and Supplementary Note 5), confirming that this signal arises from oxygen ions in close proximity to proton. Such signal is very weak in the spectrum of NO101-TiO₂ while an additional peak can be found centered at −75 ppm (Fig. 1 and Supplementary Figs. 10, 14). According to the shift, this lower-frequency resonance is assigned to adsorbed water molecules (see Supplementary Fig. 10 and Supplementary Note 3). The observation of surface OH species on the (001) facet while only water on the (101) surface, on the vacuum-dried samples, agrees with the previous DFT calculations that water prefers to dissociate on anatase TiO₂ (001) facet to form surface OHs[36], while it tends to adsorb molecularly on (101) surface[37, 38].

**Surface reconstruction of anatase TiO₂(001).** In order to help the spectral assignment, DFT calculations were performed on anatase titania structures with different exposed facets. Since water molecules prefer to dissociate on the high-energy (001) facets[36], and surface reconstructions are likely to occur on (001)[39], four possible surface models were constructed for NS001-TiO₂, including the un-reconstructed clean TiO₂(001) (CL), hydrated TiO₂(001) at a water coverage of $^1/_2$ ML

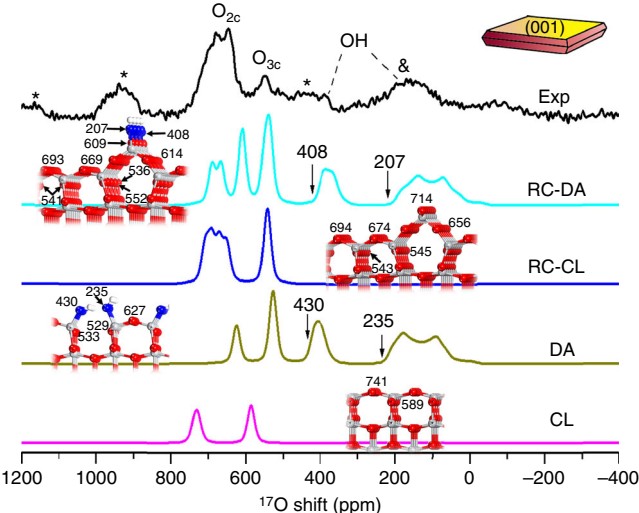

**Fig. 2** Experimental and simulated $^{17}O$ NMR spectra of NS001-TiO₂. NS001-TiO₂ (Exp) was surface selectively $^{17}O$-labeled and vacuum-dried for 2 h. The simulated spectra are based on DFT calculations on different structures, i.e., un-reconstructed clean anatase TiO₂(001) (CL), hydrated anatase TiO₂(001) at a water coverage of $^1/_2$ molecular layer (DA), 1 × 4-reconstructed clean anatase TiO₂(001) (RC-CL), and hydrated 1 × 4-reconstructed anatase TiO₂(001) (RC-DA). *Insets* are models of corresponding surface structures, where *gray*, *white*, *blue*, and *red spheres* represent Ti, H, O in surface hydroxyl groups, and other O species, respectively. Full view of all four structural models, isotropic chemical shifts of each oxygen sites and their quadrupolar parameters are presented in Supplementary Figs. 15–18 and Supplementary Tables 5–8. *Asterisks* denote sidebands. *Ampersand* denotes sideband overlapping with the OH signal. The *arrows* show the calculated isotropic chemical shift values of surface oxygen species. The centers of masses of the resonances owing to OH appear at lower frequencies due to the significant second-order quadrupolar induced shifts

(dissociative adsorption, DA), 1 × 4-reconstructed clean TiO₂(001) (RC-CL), and hydrated 1 × 4-reconstructed TiO₂(001) (RC-DA) (see Fig. 2 and Supplementary Figs. 15–18 for details). $^1/_2$ ML means that every two surface Ti$_{5c}$ take one water molecule, and it also corresponds to a fully hydrated surface state[36]. The calculated isotropic chemical shifts of each oxygen sites ($\delta_{iso}$), quadrupolar coupling constant ($C_Q$), asymmetry parameter ($\eta$), and center of gravity of the NMR signals ($\delta_{CG}$) are given in Supplementary Tables 5–8. In all the models investigated, the calculated chemical shifts ($\delta_{CG}$) of oxygen ions in the "bulk" part (middle layers) of the anatase structures are close to 558 ppm, which is the observed chemical shift of O$_{3c}$ in the nonselectively labeled anatase TiO₂. The chemical shifts of the oxygen species in the first few layers, however, deviate noticeably from the "bulk" values and depend on the specific local structure.

The calculated results were used to simulate the $^{17}O$ NMR spectra at different external magnetic fields (Fig. 2 and Supplementary Fig. 19) by considering the surface oxygen species only, whose isotropic chemical shifts have been marked in the structural models in Fig. 2. The simulated signals arising from the OH species generated in the DA and RC-DA structures give a fair match with the experimental data (450–0 ppm), further supporting that water dissociates on the (001) facets. Furthermore, they also allow us to assign a weak peak centered at approximately 400 ppm that overlaps with the sidebands from the surface oxygen sites to another OH environment. The calculation results also show that the majority species that give rise to the signals at 600–760 ppm in the experimental data are actually the O$_{2c}$ environments, rather than the O$_{3c}$ site, and that these species can only be ascribed to reconstructed surfaces (i.e., contributions from the RC-CL and/or RC-DA structures). Therefore, these results provide compelling evidence that structure reconstruction does indeed occurs on the (001) surface. On the basis of the $^1H$ NMR results (Supplementary Fig. 20 and Supplementary Table 9), the water coverage on this sample is 0.3 ML, indicating that a reconstructed surface is energetically favored at this state[36], and both RC-CL and RC-DA surface conditions should exist, due to the insufficient water coverage. Therefore, it can be concluded that, at this specific water coverage (0.3 ML), surface reconstruction occurs on (001) surface of anatase titania, and water dissociates on this surface.

**Step edges of anatase TiO₂(101).** For NO101-TiO₂, three defect-free structure models, including clean anatase TiO₂(101) (CL), hydrated anatase TiO₂(101) under a water (molecular adsorption) coverage of $^1/_2$ ML (MA), and hydrated anatase TiO₂(101) with dissociatively adsorbed water under the coverage of $^1/_2$ ML (DA, which is energetically less favorable[37, 38]), were constructed first to calculate the NMR parameters (Supplementary Figs. 21–23 and Supplementary Tables 10–12). However, the simulated spectra do not match the experimental data (for surface O$_{2c}$ sites in particular) (Supplementary Fig. 24). Surface defects, however, often occur on the (101) facets according to scanning tunneling microscopy investigations[40, 41] as well as first-principles calculations[41]. Particularly, "step edges", associated with higher reactivity[41], are considered as the most common defects on this surface. Gong et al. have proposed several types of step-edge defects[42] with monoatomic height along trapezoidal or triangular islands on (101) surface[40, 43]. The so-called type-D steps occur along two nonparallel sides of the trapezoidal islands (or two sides of the triangular ones), and they are also the most prevalent ones among all the steps. Accordingly, in the current work, an anatase TiO₂(134) vicinal surface with such type-D steps

**a**

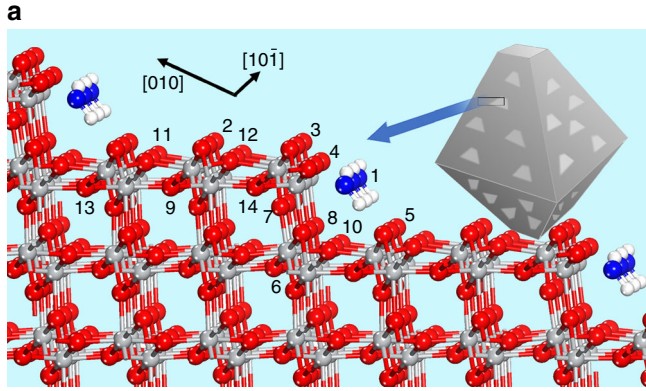

**b**

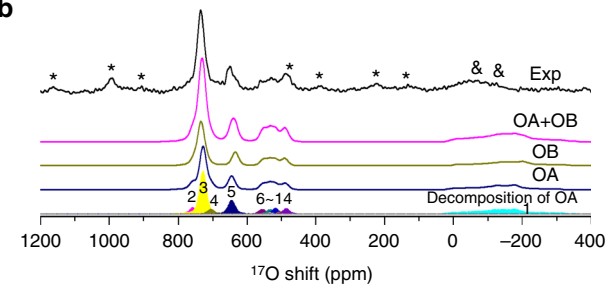

**Fig. 3** The structure model and $^{17}O$ NMR spectra of NO101-TiO$_2$. **a** The structure model of the TiO$_2$(134) vicinal surface for DFT calculations, which contains type-D steps and (101) planes. **b** The experimental $^{17}O$ spin-echo NMR spectrum of the fully dried surface-selectively $^{17}O$-labeled NO101-TiO$_2$ (*black line*) and the simulated spectra (*colored lines* and *peaks*) by using parameters obtained from DFT calculations. Water molecules are adsorbed in two orientations (OA and OB). The contributions of both adsorption orientations are also shown in **b** (*dark yellow line* for OB and *blue line* for OA). Other *colored peaks* denote the individual components of OA, which correspond to the oxygen atoms labeled with the same numbers in the structural model in **a**. The parameters adopted in the simulation are listed in Supplementary Table 16. Full views of the models with the two adsorption orientations are presented in Supplementary Figs. 25–26. *Asterisks* denote spinning sidebands, while *ampersands* denote sidebands that overlap with the signal of the adsorbed water

and (101) planes (see Fig. 3a) was constructed for the chemical shift calculations.

According to our calculations, water are molecularly adsorbed at the Ti$_{5c}$ sites (TiO$_5$) of type-D step-edges and have two different orientations (denoted as OA and OB) with similar adsorption strength, distinguished by the lengths of the hydrogen bonds formed with the adjacent oxygen ions at the edge (Supplementary Figs. 25–26 and Supplementary Tables 13–14). In both adsorption modes, water has higher adsorption energies than that found at flat (101) surface (Supplementary Table 15). Since the adsorbed water molecules in two orientations have similar adsorption energies, each orientation is weighted the same and only 14 different surface/subsurface oxygen species are considered in the spectral simulation. The calculated structures, NMR parameters, and simulated spectra, along with the experimental data, are shown in Fig. 3, Supplementary Fig. 27, and Supplementary Table 16. For clarity, the simulated spectrum of the 14 oxygen species in OA is also presented as *colored and shaded peaks* in Fig. 3b.

The simulated spectra agree remarkably well with the experimental data at different external magnetic fields (Fig. 3b and Supplementary Fig. 27b), except for the center of gravity of the NMR signal for the adsorbed water species (Fig. 3b, peak 1).

The experimental line width of this peak is smaller than the calculated one, which can be attributed to the motion of the adsorbed water molecules (see Supplementary Fig. 28 and Supplementary Note 6). Other signals from surface sites probably originate from the dissociation of H$_2$$^{17}$O at oxygen vacancies generated in the vacuum-drying pretreatment at 100 °C (see Supplementary Fig. 29 and Supplementary Note 7) and possible subsequent migration of oxygen ions within the structure of TiO$_2$, since water molecules are not expected to dissociate on type-D step edges[37, 44]. The major resonance at 730 ppm (peak 3) arises from O$_{2c}$ species at the step edges (Fig. 3). In comparison, peak 2, corresponding to O$_{2c}$ species at the middle of (101) plane, has much smaller intensity. Considering the fact that there is only a small fraction of oxygen ions at step edges (4 ± 1.5%)[42], the much stronger intensity of peak 3 implies that O$_{2c}$ at the step edge has higher activity in the initial labeling process than the species on (101) plane. The other relatively strong peak owing to O$_{2c}$ ions occurs at 640 ppm (peak 5). Such oxygen species is at flat terraces below the adjacent step edge and is attached with the adsorbed water through hydrogen bond. The signals at 480–560 ppm can be assigned to surface and subsurface O$_{3c}$ species. The much stronger intensity of the O$_{2c}$ species compared to the O$_{3c}$ ones confirms that the $^{17}O$-enrichment method adopted in this work does achieve an effective surface-selective labeling.

## Discussion

$^{17}O$ solid-state NMR spectroscopy, in combination with DFT calculations, can be used to distinguish two anatase TiO$_2$ nanocrystals with different exposed facets and explore the details of their unique surface local environments. The $^{17}O$ NMR spectra provide definitive evidence that surface reconstruction occurs when (001) faceted anatase TiO$_2$ nanosheets adsorb a small amount of water, while "step edges" are the main defects present on the anatase TiO$_2$(101) surface. The results indicate that $^{17}O$ solid-state NMR spectroscopy is a sensitive method to probe the local environments of the exposed facets of oxide nanocrystals, the structures of these facets playing a vital role in determining their properties. Further studies based on this approach can be readily envisaged to study possible changes that may occur on the faceted oxide nanocrystals in catalytic processes and other related applications.

## Methods

**Sample preparation**. The anatase TiO$_2$ nanosheets, mainly dominated by exposed (001) facets, i.e. NS001-TiO$_2$, were prepared according to Han's work[45]. (101) facets dominated anatase nano-octahedra (NO101-TiO$_2$), and non-faceted anatase TiO$_2$ nanoparticles (NF2-TiO$_2$) were prepared hydrothermally according to Liu's work[31]. The obtained materials were washed thoroughly with NaOH aqueous solution and water to remove F$^-$ or Cl$^-$ on the surface, which were introduced in the preparation. Experiment details are given in the Supplementary Methods. Another non-faceted anatase TiO$_2$ sample with smaller surface area, NF1-TiO$_2$, was purchased from Sigma-Aldrich Corporation, and used as received.

**Characterization**. The powder XRD analysis was carried out on a Philips X'Pro X-ray diffractometer using Cu Kα irradiation (λ = 1.54184 Å) operated at 40 kV and 40 mA at 25 °C. High-resolution TEM images were obtained on an FEI Titan 80/300 S/TEM with an acceleration voltage of 200 kV. Electron paramagnetic resonance (EPR) spectra were recorded on the samples with the same mass (50 mg) by a Bruker EMX-10/12 spectrometer at room temperature. The Brunauer–Emmett–Teller specific surface areas of the samples were measured by nitrogen adsorption at 77 K using a Micromeritics tristar ASAP 2020 instrument. The contents of C and N impurities of the samples were analyzed using a Heraeus CHN-0-Rapid analyzer. XPS spectra of both faceted samples were obtained on an Ulvac-PHI PHI 5000 VersaProbe instrument.

**$^{17}O$ enrichment**. Faceted NS001-TiO$_2$, NO101-TiO$_2$, and non-faceted NF2-TiO$_2$ nanocrystalline samples were surface-selectively $^{17}O$-labeled through a vacuum line

using 90% $^{17}$O-enriched $H_2O$ (Cambridge Isotope Laboratories). The sample (typically 300 mg) was first activated in a glass tube by vacuum drying at 100 °C for 1.5 h. After the sample was cooled down to room temperature, it was exposed to the saturated vapor of $^{17}$O-enriched $H_2O$ for 10 min for adequate adsorption. Then the sample was sealed in the glass tube, heated to 40 °C and kept at this temperature for 5 h to achieve an optimized $^{17}$O labeling of the surface oxygen species. The other non-faceted anatase $TiO_2$ sample NF1-$TiO_2$, with a smaller surface area, was $^{17}$O-labeled nonselectively by calcining in $^{17}O_2$ (70% $^{17}$O, Cambridge Isotope Laboratories) within a sealed glass tube at 500 °C for 12 h.

**Solid-state NMR measurement.** $^{17}$O MAS NMR spectra were measured on 9.4 and 14.1 T Bruker Avance III spectrometers using 4.0 mm MAS probes doubly tuned to $^{17}$O at 54.2 and 81.3 MHz, and $^1$H at 400.0 and 600.0 MHz, respectively. All samples were packed into rotors in a $N_2$ glove box. $^{17}$O chemical shift is referenced to $H_2O$ at 0.0 ppm.

**DFT calculations.** Spin-polarized DFT calculations were performed with the Perdew–Burke–Ernzerhof functional[46] by using the Vienna Ab initio Simulation Package (VASP)[47]. The $^{17}$O chemical shifts were calculated by using the linear response method. We used the project-augmented wave method[48] to describe the core-valence electron interactions in structure optimization, chemical shift, and electric field gradients (EFGs) calculations at a kinetic energy cutoff of 500 eV with Ti (3s, 3p, 3d, 4s), O (2s, 2p), and H (1s) electrons being treated as valence electrons. All of the atoms were allowed to relax during structure optimization with a force stopping-criterion of 0.02 eV/Å on each relaxed ion. During electronic minimization, we used an extremely high stopping criterion of $10^{-8}$ eV for all the calculations[27]. With a $3 \times 3 \times 3$ k-point mesh, we obtained optimized lattice parameters of $a = 3.80$ Å and $c = 9.51$ Å for bulk anatase $TiO_2$, which is very close to the experimental values ($a = 3.78$ Å and $c = 9.50$ Å)[49]. It should be noted that the on-site Coulomb interaction of localized d electrons was also considered by using the DFT+U approach with an optimum Hubbard U value of 4.0 eV[50], and lattice parameters of $a = 3.86$ Å and $c = 9.53$ Å was obtained. This indicates pristine DFT method can give reliable structural information. Since correct structural information is crucial to chemical shift calculations, we then used the pristine DFT method to do all the calculations.

The anatase $TiO_2$ structures were modeled by surface slabs that are thick enough to maintain trivial fluctuations of chemical shift values in their middle layers (see Supplementary Figs. 15–18 and 21–23 for details). For un-reconstructed $TiO_2$(001) surface, $1 \times 4$-reconstructed $TiO_2$(001) surface[39], $TiO_2$(101) surface, and $TiO_2$(134) vicinal surface consisting of type-D steps and (101) planes[42], we used a $1 \times 2$, $2 \times 4$, $1 \times 2$, and $1 \times 1$ surface cell, respectively, with a corresponding $4 \times 2 \times 1$, $2 \times 1 \times 1$, $2 \times 2 \times 1$, and $2 \times 3 \times 1$ k-point mesh, respectively, for the Brillouin zone integration. All the slabs also contain a large vacuum gap (~12 Å for un-reconstructed anatase $TiO_2$(001), $1 \times 4$-reconstructed anatase $TiO_2$(001), anatase $TiO_2$(101) surfaces, and ~13 Å for anatase $TiO_2$(134) vicinal surfaces) to remove the slab–slab interactions.

The isotropic chemical shift ($\delta_{iso}$) can be computed as $\delta_{iso} = \delta_{cal} + \delta_{ref}$[27], where $\delta_{cal}$ is the chemical shift obtained in VASP, $\delta_{ref}$ is the reference chemical shift. Considering the fact that bulk oxygens have more regular arrangements than those near the surfaces, all the $\delta_{ref}$ for each model (except $TiO_2$(134) vicinal surface) were determined by aligning the average $\delta_{cal}$ of middle four layers to the experimental $\delta_{iso}$ of bulk $O_{3c}$ (561 ppm, Supplementary Figs. 15–18, 21–23). For the anatase $TiO_2$(134) vicinal surface consisting of type-D steps and (101) planes, shown in Supplementary Figs. 25 and 26, the average $\delta_{iso}$ of atom 35–39, which are in the middle layer, is set as $\delta_{iso}$ of bulk $O_{3c}$. All determined $\delta_{ref}$ is in the range of 50–60 ppm (given in caption of Supplementary Figs. 15–18, 21–23, 25, 26), which is close to the reported value of 52 ppm for $CeO_2$[27].

To calculate the quadrupole coupling constant ($C_Q$) and asymmetry parameter ($\eta$), we used the following equations:

$$C_Q = \frac{eQV_{ZZ}}{h} \tag{1}$$

$$\eta = \frac{V_{XX} - V_{YY}}{V_{ZZ}}, \tag{2}$$

where $h$ is the Planck constant, $e$ is the absolute value of the electron charge, and $V_{ii}$ (ii = XX, YY, or ZZ) are the eigenvalues of the EFG tensor with $|V_{ZZ}| > |V_{YY}| > |V_{XX}|$. We used the experimental quadrupole moment ($Q$) of −0.02558 barns[51] for $^{17}$O.

The adsorption energy of $H_2O$ ($E_{ads}$) was calculated as follows:

$$E_{ads} = E_{H_2O} + E_{sub} - E_{H_2O/sub}, \tag{3}$$

where $E_{H_2O}$, $E_{sub}$, and $E_{H_2O/sub}$ are the DFT total energies of the gas phase $H_2O$, the $TiO_2$ substrate, and the adsorption complex, respectively.

**$^{17}$O NMR spectra simulation.** Wsolids package developed by Dr. K. Eichele was used to simulate the $^{17}$O NMR spectra using the NMR parameters obtained with DFT calculations, as shown in Figs. 2, 3b and Supplementary Figs. 19b, 24, 27b, 28.

For the simulated spectra from the models of (001) facet (Fig. 2 and Supplementary Fig. 19b), only surface oxygen sites were considered, whose isotropic chemical shifts have been marked in the structural models in Fig. 2. The $O_{2c}$ and $O_{3c}$ sites have been given the same weight of peak area in the calculated spectra. Twice of the weight has been given to the hydroxyl groups centered around 420 ppm, and four times of the weight has been given to the hydroxyl groups centered around 150 ppm, for the sake of presentation. For simulating the NMR spectra of the defect-free (001) facet (as shown in Supplementary Fig. 24), a similar approach was used. Surface sites, i.e., sites 1–3 in Supplementary Fig. 21 and sites 1–7 in Supplementary Figs. 22 and 23, respectively, were considered. Twice the weight of the peak areas have been given to the signals of hydroxyl groups and adsorbed water, in comparison to those of the surface $O_{2c}$ and $O_{3c}$ sites. For simulating the spectra of (101) facet with type-D steps (Fig. 3b and Supplementary Fig. 27b), NMR parameters of surface and subsurface oxygen sites (1–14) in Supplementary Tables 13 and 14 were adopted, only with their percentage adjustable to achieve the best fitting. Furthermore, in Supplementary Fig. 28, $C_Q$s of the adsorbed water in both adsorption orientations were also allowed to change in the simulation, in order to examine the influence of the motion of the adsorbed water on its NMR signal.

**Data availability.** All relevant data are available from the authors.

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

## Acknowledgements

This work was supported by the National Basic Research Program of China (2013CB934800), the National Natural Science Foundation of China (NSFC) (21573103, 21421004, 21222302, and 20903056), NSFC—Royal Society Joint Program (21661130149 and 21111130201), Program for New Century Excellent Talents in University (NCET-10-0483), the Fundamental Research Funds for the Central Universities (1124020512), and National Science Fund for Talent Training in Basic Science (J1103310). The ECUST group also thanks the Programme of Introducing Talents of Discipline to Universities (B16017) and National Super Computing Centre in Jinan for computing time. L.P. thanks Royal Society and Newton Fund for Royal Society—Newton Advanced Fellowship. C.P.G. thanks the European Research Council for an Advanced Fellowship. This work was also supported by a Project Funded by the Priority Academic Program Development of Jiangsu Higher Education Institutions.

## Author contributions

N.J., F.G. and L.D. carried out the synthesis of anatase nanostructures. Y.L., L.S., H.S., Y.W., M.W., X.G., W.H. and W.D. carried out XRD, EPR, C and N element analyzing, XPS and surface area measurement; M.L. performed HRTEM; Y.L., L.S., H.S., M.W., X.K. and L.P. performed $^{17}$O isotope enrichment, collected, and analyzed the NMR spectra; Y.L., Z.Y., X.K. and L.P. collected and analyzed the high-field NMR spectra; X.-P.W. and X.-Q.G. conducted the DFT calculations; Y.L., X.-P.W., M.L., L.D., W.H., W.D., C.P.G., X.-Q.G. and L.P. wrote the manuscript, and all authors discussed the experiments and final manuscript.

## Additional information

**Competing interests:** The authors declare no competing financial interests.

