## [Peer Review File · Nature Communications]

Reviewers' comments:

Reviewer #1 (Remarks to the Author):

This manuscript reports an extension of a previous study by the same authors (Ref. 27) on the application of ^{17}O solid-state NMR spectroscopy to identify different oxygen species on oxide nanostructures. Here the authors show that the ^{17}O NMR spectra of anatase TiO_2 nanoparticles (NPs) with dominant $\{001\}$ and $\{101\}$ facets exhibit distinct features, which are also different from those of non-faceted NPs. By comparing experimental data to DFT calculations, they further conclude that the anatase $\{001\}$ facets are reconstructed and dissociate water while $\{101\}$ facets contain a large number of steps and adsorb water in molecular form.

Altogether, I find this study interesting, well presented, and valuable, but neither the experimental and theoretical methods (already reported in previous studies) nor the specific results on anatase NPs seem to me completely new and original. In addition, I believe there are a few points that require further discussion.

- The spectrum of NO101- TiO_2 was modeled using a (134) vicinal surface, which has a quite high density of steps and therefore a significant surface energy. Is such a high step density essential to reproduce the spectrum? What would happen if another vicinal, e.g. (156), were used?
- On a related note, spectra were simulated using models that included various amounts of adsorbed water. The amount of water that is present on a surface under given conditions could be actually estimated through ab-initio thermodynamic calculations. It would be interesting to compare such an estimate with the corresponding value inferred from experiment.
- Some discussion on the reproducibility of the spectra would be desirable. For example, Figure S7 shows that all ^{17}O signals from NS001- TiO_2 decrease significantly during the second room-temperature vacuum-drying process whereas no such decrease is observed for the NO101- TiO_2 sample in Figure S8. Is this difference typical or accidental?
- The sizes of the NPs in the different TiO_2 samples used in this study are rather different. In particular the non-faceted TiO_2 particles are much larger than the NPs in the NS001- TiO_2 and NO101- TiO_2 samples (Table S2). Smaller non-faceted NPs would likely exhibit surface features that would make the comparison to NS001- TiO_2 and NO101- TiO_2 much less straightforward. Some discussion of these issues would be desirable.

A detail:

The most appropriate reference for the reconstructed anatase (001) surface is the original work by Lazzeri et al (PRL 2001, 87, 266105) rather than Ref. 36.

Reviewer #2 (Remarks to the Author):

This article is a follow-up article of the one previously published in 2015 in Sci. Adv. (DOI: 10.1126/sciadv.1400133). The authors prove here that ^{17}O NMR, especially thanks to the large chemical shift range, allow distinguishing between different surface oxygen species. The main finding, as far as I am concerned, is about how the authors treated the case of NO101- TiO_2 . In particular, they have proposed a model with step-edges that better reproduce the local surface structure of the system. The novelty in the present paper is to prove that this local structure is perfectly in line with the experimental ^{17}O NMR data. The 14 different surface/subsurface oxygen species are characterized by chemical shift spread over more than 250-300 ppm. It proves the high sensitivity of ^{17}O NMR to

local structure.

The article is clear, easy to follow, and very well written. The connexion between NMR results, DFT calculations and surface topology is well presented. The bibliographic references cited are complete and up to date for both the field of oxide nanocrystals and ^{17}O solid-state NMR.

There are a few major points to address before accepting the article.

Major Corrections

There is a concern about the comparison of nonselectively ^{17}O labeled TiO_2 with surface labeled NS001 and NO101 TiO_2 samples. Indeed, ^{17}O spectrum (Fig. 1-bottom) was acquired with single pulse (pulse length of 0.2 μs only !?). Why not use the same echo sequence than for surface labeled NS001 and NO101 TiO_2 samples in order to favour the observation of broad NMR signals, if any ? It seems to me that the NMR tools are not identical which prevents clear comparison of the bottom spectrum with the 2 top ones. What is the number of scans used for the bottom spectrum. It is not mentioned in the text.

The ^{17}O NMR spectra require very long acquisition times (here more 110000 repetitions). Is the recycling delay really optimized (how ?) or the authors just took a short one that would allow spectral acquisition in a reasonable time ?

Concerning the DFT calculations, the authors mentioned a large vacuum gap ($> 10 \text{ \AA}$) to remove the slab-slab interactions. What is "bigger than 10 \AA " ? How did they test that it was large enough ?

Minor corrections / suggestions

Even in the Supp Info, I prefer when Tables and Figures appear as they are mentioned in the text. Here the Tables are at the end of the Supp Info which make it more difficult to follow.

Reviewer #1 (Remarks to the Author):

This manuscript reports an extension of a previous study by the same authors (Ref. 27) on the application of ^{17}O solid-state NMR spectroscopy to identify different oxygen species on oxide nanostructures. Here the authors show that the ^{17}O NMR spectra of anatase TiO_2 nanoparticles (NPs) with dominant $\{001\}$ and $\{101\}$ facets exhibit distinct features, which are also different from those of non-faceted NPs. By comparing experimental data to DFT calculations, they further conclude that the anatase $\{001\}$ facets are reconstructed and dissociate water while $\{101\}$ facets contain a large number of steps and adsorb water in molecular form.

Altogether, I find this study interesting, well presented, and valuable, but neither the experimental and theoretical methods (already reported in previous studies) nor the specific results on anatase NPs seem to me completely new and original. In addition, I believe there are a few points that require further discussion.

- We thank the reviewer for the positive comments, efforts and time spent for refereeing our paper. In this work, we successfully distinguished different exposed facets of anatase TiO_2 nanocrystals by using ^{17}O solid-state NMR spectroscopy, for the first time. Since the exposed facets of nanocrystals play a vital role in controlling the properties (e.g., catalytic properties), we believe our NMR approach, which is based on the behaviors of an ensemble of nuclei in the sample and thus more representative, provides an important alternative to electron microscopy techniques. Furthermore, we show that ^{17}O NMR spectra of these nanocrystals with certain exposed facets reflect fine details of their surface structure. As the reviewer has mentioned, our results show anatase $\{001\}$ facets are reconstructed while $\{101\}$ facets contain a number of steps. Such detailed information can hardly be obtained with other spectroscopic methods.
- The spectrum of NO101- TiO_2 was modeled using a (134) vicinal surface, which has a quite high density of steps and therefore a significant surface energy. Is such a high step density essential to reproduce the spectrum? What would happen if another vicinal, e.g. (156), were used?
- We thank the reviewer for raising his or her concern on the effect of step density. To address this concern, we computed the isotropic chemical shifts and quadrupolar parameters for oxygen ions in a hydrated anatase $\text{TiO}_2(156)$ vicinal surface model with a decreased step density (see Figure X1 in this letter for the structure, and the corresponding structure for hydrated anatase $\text{TiO}_2(134)$ can be found in Fig. 3 in the manuscript and Fig. S24 in the Supplementary Information), as suggested by the reviewer. The calculated isotropic chemical shifts and the quadrupolar parameters are shown in Table X1 (see Table S13 in the Supplementary Information, for hydrated $\text{TiO}_2(134)$). It is clear the NMR parameters of the oxygen ions close to steps (e.g., O1-O5) do not change much with the decrease of step density, which suggests that step-step interactions are trivial in this simulation and the stepped surface model (hydrated anatase $\text{TiO}_2(134)$) used in our work is reliable to reproduce the spectrum. The simulated spectra for the two models (hydrated anatase (134) and (156)) are almost identical as well (Figure X2). Although the contributions from O101/102 and O103/104, which are oxygen ions further away from the steps, are not

counted in the simulated spectrum for the hydrated anatase $\text{TiO}_2(156)$ model, their NMR parameters are very similar to O2 and O11 (oxygen ions closer to the steps and are included in the simulation for both hydrated anatase $\text{TiO}_2(134)$ and (156) models), respectively. These results indicate the contributions from O101/102 and O103/104 are well represented by O2 and O11, and a larger vicinal surface model such as hydrated anatase $\text{TiO}_2(156)$ which will cost much longer computing time is not necessary in this case.

Figure X1. Calculated structure of hydrated anatase $\text{TiO}_2(156)$ vicinal surface (one water molecule adsorbs at the step edge with OA orientation). Oxygen ions close to the steps are numbered from 1 as the way in the model of $\text{TiO}_2(134)$ vicinal surface. Other oxygen ions further away from the steps are numbered from 101.

Table X1. Calculated isotropic chemical shifts (δ_{iso}), quadrupolar parameters (C_Q and η) and center of gravity of the NMR signals (δ_{CG}) for oxygen ions in hydrated anatase $\text{TiO}_2(156)$ (one water molecule adsorbs at the step edge with OA orientation). The corresponding oxygen ions are labeled in Figure X1. $\delta_{\text{ref}} = 51$.

Oxygen sites	$\delta_{\text{iso}}/\text{ppm}$	C_Q/MHz	η	$\delta_{\text{CG}}/\text{ppm}$
1	16	8.51	0.70	-156
2	763	1.24	0.71	759
3	727	1.08	0.98	724
4	708	0.57	0.95	707
5	655	1.72	0.26	649
6	559	1.31	0.57	555
7	552	1.34	0.92	547
8	549	1.59	0.14	544
9	536	1.44	0.93	531

10	521	1.39	0.59	517
11	501	1.14	0.94	498
12	489	1.26	0.90	485
13	538	1.49	0.66	533
14	549	1.32	0.85	545
15	551	1.21	0.62	548
16	554	1.13	0.99	551
17	555	1.38	0.57	551
18	552	1.87	0.38	545
19	549	1.23	0.50	546
20	551	1.22	0.40	548
21	571	1.69	0.27	565
22	563	1.14	0.43	560
23	565	1.39	0.49	561
24	564	1.04	0.82	561
25	563	1.35	0.46	559
26	558	1.24	0.41	555
27	560	1.23	0.35	557
28	560	1.25	0.66	556
29	560	1.23	0.47	557
30	560	1.41	0.27	556
31	561	1.08	0.76	558
32	559	1.25	0.33	556
33	560	1.45	0.35	556
34	562	1.30	0.48	558
35	560	1.22	0.38	557
36	561	1.25	0.45	558
37	561	1.24	0.38	558
38	561	1.25	0.33	558
39	562	1.24	0.50	559
101	766	1.31	0.71	762
102	766	1.28	0.71	762
103	501	1.30	0.80	497
104	500	1.27	0.84	496
105	534	1.47	0.70	529
106	555	1.09	0.93	552
107	534	1.47	0.94	528
108	555	1.09	0.99	552
109	558	1.79	0.29	551
110	546	1.22	0.47	543
111	562	1.74	0.28	556
112	548	1.27	0.45	544

113	560	1.19	0.61	557
114	563	1.08	0.82	560
115	562	1.21	0.59	559
116	563	1.05	0.79	560
117	559	1.44	0.36	555
118	558	1.24	0.38	555
119	560	1.43	0.34	556
120	558	1.24	0.41	555
121	559	1.22	0.41	556
122	562	1.26	0.46	559
123	560	1.24	0.41	557
124	562	1.26	0.46	559
125	562	1.25	0.47	559
126	560	1.24	0.41	557
127	562	1.25	0.46	559
128	560	1.22	0.41	557

Figure X2. Simulated ^{17}O NMR spectra for hydrated anatase TiO_2 (134) and (156) (one water molecule adsorbs at the step edge with OA orientation). The oxygen ions selected for simulation (i.e. oxygen ions 1-14 in Figure 3 of the manuscript and Figure X1) are labeled in blue.

- On a related note, spectra were simulated using models that included various amounts of adsorbed water. The amount of water that is present on a surface under given conditions could be actually estimated through ab-initio thermodynamic calculations. It would be interesting to compare such an estimate with the corresponding value inferred from experiment.
- We agree with the reviewer that the amount of water on a surface under given conditions could be estimated by *ab initio* thermodynamic calculations. However, locating the most stable structures at each specific water coverage, which should be done first in such thermodynamic calculations, could be a huge work. Furthermore, for large chemical systems, such as the models we used for this work, where models include several hundred atoms, the “chemical accuracy” in terms of energies required for precise thermodynamic calculations is hard to achieve. In this manuscript, we used quantitative ^1H NMR measurements (spin counting) to determine the water content on the surface of TiO_2 nanoparticles, by using adamantane as the standard material. This is a convenient and standard method, which proves to be reliable to determine the total as well as different individual H species (e.g., Geoffrey Hartmeyer *et al.*, *J. Phys. Chem. C*, **2007**, *111*, 9066). For example, the water coverage on NS001- TiO_2 sample determined by ^1H NMR is 0.3 ML. Therefore, two representative models, i.e., clean anatase $\text{TiO}_2(001)$ and hydrated anatase $\text{TiO}_2(001)$ at a water coverage of 0.5 ML were first constructed and calculated. Considering that surface reconstruction of anatase $\text{TiO}_2(001)$ could occur at this experimental condition, we also constructed 1 \times 4-reconstructed clean anatase $\text{TiO}_2(001)$ and hydrated 1 \times 4-reconstructed anatase $\text{TiO}_2(001)$ models. By combining the calculation and simulation results from these four representative models, the NMR spectrum of NS001- TiO_2 can be appropriately analyzed.
- Some discussion on the reproducibility of the spectra would be desirable. For example, Figure S7 shows that all 17O signals from NS001- TiO_2 decrease significantly during the second room-temperature vacuum-drying process whereas no such decrease is observed for the NO101- TiO_2 sample in Figure S8. Is this difference typical or accidental?
- All of the experiments were repeated several times and very similar spectra were obtained. The intensities of the signals of NS001- TiO_2 show some decrease while such decrease is not significant for NO101- TiO_2 . The decrease of all ^{17}O NMR signals from NS001- TiO_2 may arise from a variety of processes during the second room temperature vacuum drying, including the dehydroxylation of surface OH group, possible dynamic exchange of different surface sites (i.e., non-hydroxyl surface oxygen ions with hydroxyl groups), as well as isotopic exchange between surface oxygen ions and a small amount of unlabeled water that may enter the vacuum tube. The observation that the signals of NO101- TiO_2 do not decrease much during the prolonged vacuum drying process may be associated with the lower activity of the (101) facet or smaller hydroxyl concentration. Related discussion has been added to the Supplementary Information on Pages 17-18.

- The sizes of the NPs in the different TiO₂ samples used in this study are rather different. In particular the non-faceted TiO₂ particles are much larger than the NPs in the NS001-TiO₂ and NO101-TiO₂ samples (Table S2). Smaller non-faceted NPs would likely exhibit surface features that would make the comparison to NS001-TiO₂ and NO101-TiO₂ much less straightforward. Some discussion of these issues would be desirable.
- We thank the reviewer for this helpful suggestion. The larger non-faceted TiO₂ particles were non-selectively labeled and used mostly to show the ¹⁷O shift of oxygen ions in the “bulk” part of anatase. We also agree with the reviewer that smaller non-faceted anatase TiO₂ nanoparticles should be a good comparison. Therefore, we have now added the comparison of the ¹⁷O NMR spectra of the faceted nanocrystals to a smaller non-faceted nanoparticle sample (NF2-TiO₂) to the Supplementary Information, (Supplementary Fig. S11). The smaller non-faceted nanoparticles have an average diameter of about 10 nm (Supplementary Fig. S10), thus more comparable to NS001-TiO₂ and NO101-TiO₂ samples. The ¹⁷O NMR spectrum of smaller non-faceted nanoparticles show much broader and rather featureless resonances due to 2-coordinated oxygen ions (O_{2c}) on the surface, as compared to faceted samples NS001-TiO₂ and NO101-TiO₂, indicating more complicated surface environments originated from the presence of different facets. The related discussion has been added to the Supplementary Information after Fig. S11.

A detail:

The most appropriate reference for the reconstructed anatase (001) surface is the original work by Lazzeri et al (PRL 2001, 87, 266105) rather than Ref. 36.

- We thank the reviewer for the helpful suggestion. Now we have added this reference (ref. 39). We have also replaced reference 36 with J. Phys. Chem. B 2005, 109, 19560, which is the original work for the behavior of water on TiO₂(001).

Reviewer #2 (Remarks to the Author):

This article is a follow-up article of the one previously published in 2015 in Sci. Adv. (DOI: 10.1126/sciadv.1400133). The authors prove here that ¹⁷O NMR, especially thanks to the large chemical shift range, allow distinguishing between different surface oxygen species. The main finding, as far as I am concerned, is about how the authors treated the case of NO101-TiO₂. In particular, they have proposed a model with step-edges that better reproduce the local surface structure of the system. The novelty in the present paper is to prove that this local structure is perfectly in line with the experimental ¹⁷O NMR data. The 14 different surface/subsurface oxygen species are characterized by chemical shift spread over more than 250-300 ppm. It proves the high sensitivity of ¹⁷O NMR to local structure.

The article is clear, easy to follow, and very well written. The connexion between NMR results, DFT calculations and surface topology is well presented. The bibliographic references cited are complete and up to date for both the field of oxide nanocrystals and ^{17}O solid-state NMR.

- We would like to thank the reviewer for the support and for all the comments provided below.

Major Corrections

There is a concern about the comparison of nonselectively ^{17}O labeled TiO_2 with surface labeled NS001 and NO101 TiO_2 samples. Indeed, ^{17}O spectrum (Fig. 1-bottom) was acquired with single pulse (pulse length of 0.2 μs only !?). Why not use the same echo sequence than for surface labeled NS001 and NO101 TiO_2 samples in order to favour the observation of broad NMR signals, if any ? It seems to me that the NMR tools are not identical which prevents clear comparison of the bottom spectrum with the 2 top ones. What is the number of scans used for the bottom spectrum. It is not mentioned in the text.

- We thank the reviewer for the helpful suggestions. We have acquired the spectrum of the nonselectively ^{17}O -labeled TiO_2 sample (now denoted as NF1- TiO_2 to differentiate from the much smaller non-faceted TiO_2 nanoparticles, NF2- TiO_2) using the same rotor synchronized Hahn-echo (Spin-echo) sequence for NS001- TiO_2 and NO101- TiO_2 samples (Figure 1). The resonant frequency and the line width of the sharp peak for the nonselectively ^{17}O -labeled TiO_2 are not changed. We have also specified the number of scans used for all the spectra in the figure caption of Figure 1. The details of the experimental parameters including the time used to acquire the spectra at 9.4 T are also summarized in Supplementary Table S4.

The ^{17}O NMR spectra require very long acquisition times (here more 110000 repetitions). Is the recycling delay really optimized (how ?) or the authors just took a short one that would allow spectral acquisition in a reasonable time ?

- We did optimize the recycle delay time for the long Hahn-echo (Spin-echo) NMR experiments. In order to achieve a reasonable S/N within a short period of time, we used a standard single pulse sequence (with ^1H decoupling) instead of Hahn-echo sequence. A relatively small acquisition number (800) was used in these optimization experiments and a set of spectra with different recycle delays were recorded (the spectra with a recycle delay of 0.2, 0.5 and 1.0 s are shown in Supplementary Fig. S7). Despite the wavy baseline due to the use of single pulse sequence, it is clear that a recycle delay of 0.5 s was long enough for quantitative measurement of all the resonances, for both NS001- TiO_2 and NO101- TiO_2 . Therefore, 0.5 s was chosen to obtain the NMR spectra with a much larger acquisition number.

Concerning the DFT calculations, the authors mentioned a large vacuum gap ($> 10 \text{ \AA}$) to remove the slab-slab interactions. What is "bigger than 10 \AA " ? How did they test that it was large enough ?

- We thank the reviewer for pointing out this issue. For un-reconstructed anatase $\text{TiO}_2(001)$, 1×4 -reconstructed anatase $\text{TiO}_2(001)$, anatase $\text{TiO}_2(101)$ and anatase $\text{TiO}_2(134)$ vicinal surface models, we used nearly identical vacuum gaps for them ($\sim 12 \text{ \AA}$ for the former three surfaces, and $\sim 13 \text{ \AA}$ for anatase $\text{TiO}_2(134)$ vicinal surface). Since the hydrated anatase $\text{TiO}_2(134)$ vicinal surface model (see Figure 3 in the manuscript) is one of the most important cases in this work, we constructed a new slab of this model by just increasing 10 \AA vacuum gap from the original one ($13 \rightarrow 23 \text{ \AA}$) to study the size effect of vacuum gap. It can be seen from Table X2 that increasing the vacuum gap does not really change the isotropic chemical shifts and quadrupolar parameters for different oxygen ions. The simulated spectra for the two slabs are almost identical as well (see Figure X3). We also made corrections on Page 19 in the revised manuscript: "All the slabs also contain a large vacuum gap ($\sim 12 \text{ \AA}$ for un-reconstructed anatase $\text{TiO}_2(001)$, 1×4 -reconstructed anatase $\text{TiO}_2(001)$, anatase $\text{TiO}_2(101)$ surfaces, and $\sim 13 \text{ \AA}$ for anatase $\text{TiO}_2(134)$ vicinal surfaces) to remove the slab-slab interactions".

Table X2. The calculated isotropic chemical shifts and quadrupolar parameters of oxygen ions on the $\text{TiO}_2(134)$ vicinal surfaces using two different vacuum gaps, and the absolute differences of these NMR parameters between the two.

O Site	13 \AA			23 \AA			Difference		
	$\delta_{\text{iso}}/\text{ppm}$	C_Q/MHz	η	$\delta_{\text{iso}}/\text{ppm}$	C_Q/MHz	η	$\delta_{\text{iso}}/\text{ppm}$	C_Q/MHz	η
1	21	8.37	0.71	18	8.42	0.71	3	0.05	0.00
2	761	1.27	0.71	760	1.27	0.71	1	0.00	0.00
3	730	1.09	1.00	728	1.09	1.00	2	0.00	0.00
4	705	0.59	0.81	706	0.57	0.89	1	0.02	0.08
5	650	1.74	0.24	649	1.74	0.24	1	0.00	0.00
6	558	1.28	0.55	557	1.28	0.55	1	0.00	0.00
7	552	1.33	0.95	550	1.33	0.96	2	0.00	0.01
8	547	1.60	0.13	547	1.61	0.13	0	0.01	0.00
9	536	1.38	0.81	535	1.38	0.81	1	0.00	0.00
10	520	1.39	0.57	519	1.40	0.57	1	0.01	0.00
11	499	1.12	0.97	499	1.12	0.97	0	0.00	0.00
12	488	1.21	0.93	487	1.22	0.93	1	0.01	0.00
13	537	1.48	0.68	537	1.48	0.68	0	0.00	0.00
14	548	1.31	0.87	547	1.31	0.87	1	0.00	0.00
15	549	1.23	0.63	548	1.23	0.62	1	0.00	0.01
16	552	1.13	0.96	551	1.12	0.97	1	0.01	0.01
17	554	1.37	0.56	554	1.37	0.55	0	0.00	0.01
18	553	1.85	0.39	553	1.86	0.39	0	0.01	0.00
19	552	1.27	0.47	551	1.27	0.46	1	0.00	0.01
20	552	1.23	0.39	551	1.24	0.39	1	0.01	0.00

21	567	1.72	0.29	566	1.73	0.30	1	0.01	0.01
22	562	1.16	0.42	562	1.16	0.43	0	0.00	0.01
23	565	1.39	0.51	564	1.39	0.50	1	0.00	0.01
24	564	1.05	0.84	564	1.06	0.84	0	0.01	0.01
25	563	1.34	0.46	563	1.34	0.45	0	0.00	0.01
26	559	1.23	0.43	558	1.23	0.42	1	0.00	0.01
27	560	1.23	0.38	560	1.23	0.37	0	0.00	0.01
28	559	1.22	0.67	559	1.22	0.68	0	0.00	0.01
29	562	1.25	0.50	561	1.25	0.49	1	0.00	0.01
30	559	1.41	0.27	558	1.41	0.26	1	0.00	0.01
31	561	1.05	0.75	561	1.06	0.75	0	0.01	0.00
32	560	1.26	0.33	560	1.26	0.32	0	0.00	0.01
33	559	1.44	0.32	559	1.44	0.31	0	0.00	0.01
34	562	1.29	0.48	561	1.30	0.48	1	0.01	0.00
35	561	1.22	0.40	560	1.22	0.40	1	0.00	0.00
36	562	1.24	0.43	561	1.25	0.43	1	0.01	0.00
37	560	1.24	0.40	560	1.24	0.40	0	0.00	0.00
38	562	1.26	0.33	561	1.27	0.32	1	0.01	0.01
39	562	1.23	0.47	561	1.24	0.46	1	0.01	0.01

Figure X3. Simulated ^{17}O NMR spectra for hydrated anatase $\text{TiO}_2(134)$ slabs (one water molecule adsorbs at each two step-edge Ti_{5c} ions with OA orientation) with vacuum gaps of 13 and 23 Å. The oxygen ions selected for simulation are oxygen ions 1-14 in Figure 3 in the manuscript.

Minor corrections / suggestions

Even in the Supp Info, I prefer when Tables and Figures appear as they are mentioned in the text. Here the Tables are at the end of the Supp Info which make it more difficult to follow.

- We thank the reviewer for this suggestion. We have revised the Supplementary Information and put the Figures and Tables in the order that the reviewer suggested.

REVIEWERS' COMMENTS:

Reviewer #1 (Remarks to the Author):

I have reviewed the response files. The authors have addressed the comments from the previous reviewers carefully. I am satisfied with their responses and the changes in the manuscript and supporting material. It is a work of very good quality that confirms the value of ^{17}O NMR spectroscopy for the study of oxide nanostructures.

Reviewer #2 (Remarks to the Author):

The article can now be published as such. Thank you for consideration my suggestions and for clarifying a few points.